# Maximum Tolerated Dose and Anti-Tumor Activity of Intraperitoneal Cantrixil (TRX-E-002-1) in Patients with Persistent or Recurrent Ovarian Cancer, Fallopian Tube Cancer, or Primary Peritoneal Cancer: Phase I Study Results

**DOI:** 10.3390/cancers13133196

**Published:** 2021-06-26

**Authors:** Jermaine I. Coward, Minal A. Barve, Ganessan Kichenadasse, Kathleen N. Moore, Paul R. Harnett, Daniel Berg, James S. Garner, Don S. Dizon

**Affiliations:** 1ICON Cancer Centre, South Brisbane, QLD 4101, Australia; 2School of Medicine, University of Queensland, Brisbane, QLD 4072, Australia; 3Mary Crowley Cancer Research Center, Dallas, TX 75251, USA; mbarve@marycrowley.org; 4Flinders Medical Centre, Adelaide, SA 5042, Australia; ganessan.kichenadasse@sa.gov.au; 5Department of Gynecologic Oncology, University of Oklahoma Health Science Center, Stephenson Cancer Center, Oklahoma City, OK 73104, USA; kathleen-moore@ouhsc.edu; 6Crown Princess Mary Cancer Centre, Westmead, NSW 2145, Australia; paul.harnett@health.nsw.gov.au; 7Formerly of Kazia Therapeutics Ltd., Sydney, NSW 2000, Australia; daniel.berg6558@gmail.com; 8Kazia Therapeutics Ltd., Sydney, NSW 2000, Australia; james.garner@kaziatherapeutics.com; 9Lifespan Cancer Institute, Providence, RI 02913, USA; ddizon@lifespan.org; 10The Warren Alpert Medical School, Brown University, Providence, RI 02912, USA

**Keywords:** ovarian cancer, platinum refractory, platinum resistant, intraperitoneal delivery, phase I

## Abstract

**Simple Summary:**

Survival outcomes with standard cytotoxic chemotherapy are poor, and most patients with ovarian cancer will die with platinum-resistant disease. This may reflect the existence of drug-resistant ovarian cancer stem cells. Cantrixil is a novel third-generation benzopyran molecule, with potent cytotoxicity against chemoresistant ovarian cancer stem cells and chemosensitive ovarian cancer cell lines. The aims of this Phase I study were to define the maximum tolerated dose, tolerability, and antitumor activity of Cantrixil when administered via an intraperitoneal (IP) port. Cantrixil was tolerable even in patients with heavily pretreated disease. This first in-human study has demonstrated the potential for prolonged survival in advanced ovarian cancer by inducing ovarian cancer stem cells’ death and sensitizing cells to standard chemotherapy with IP-administered Cantrixil. Future studies should focus on confirming its mechanism of action alongside further assessment of cancer stem cell biomarkers, and determining optimal clinical settings to maximize survival outcomes in ovarian cancer.

**Abstract:**

Survival outcomes in ovarian cancer are poor. The aims of this Phase I progressive design study (NCT02903771) were to evaluate the maximum tolerated dose (MTD), tolerability, and antitumor activity of Cantrixil—a novel third-generation benzopyran molecule—in patients (*n* = 25) with advanced, recurrent/persistent epithelial ovarian, primary peritoneal, or fallopian tube cancer. All had completed ≥ 2 prior regimens; 3 (12%) had platinum-refractory disease, and 17 (68%) had platinum-resistant disease. Following intraperitoneal (IP) port placement, patients received weekly IP Cantrixil in 3-week cycles as monotherapy (Cycles 1–2), and then in combination with intravenous (IV) chemotherapy (Cycles 3–8). Part A (dose escalation) enrolled 11 patients in 6 dose-level cohorts. An MTD of 5 mg/kg was established with dose-limiting toxicity of ileus. Most treatment-related adverse events were gastrointestinal. Across Parts A and B (dose expansion), 16 (64%) patients received ≥ 1 3-week Cantrixil cycle, and had ≥ 1 post-baseline efficacy measurement available. The results show promising anti-tumor activity in monotherapy (stable disease rate of 56%) and in combination with IV chemotherapy (objective response rate of 19%, disease control rate of 56%, and median progression-free survival of 13.1 weeks). The molecular target and mechanism of action of Cantrixil are yet to be confirmed. Preliminary analysis of stem cell markers suggests that IP Cantrixil might induce ovarian cancer stem cell death and sensitize cells to standard chemotherapy, warranting further evaluation.

## 1. Introduction

Each year over 300,000 people are diagnosed with ovarian cancer [1]. Approximately two-thirds of patients present with advanced disease at diagnosis [2], primarily because early disease is difficult to detect, and is hampered by the non-specific nature of the presenting symptoms and the limitations of current clinical tests [3]. The standard of care for advanced ovarian cancer comprises primary debulking followed by combination chemotherapy using paclitaxel and carboplatin [4,5]. Despite an initial clinical response, approximately 80% of patients experience disease recurrence within 2 years [6]. Treatment approaches vary, and include the use of neoadjuvant chemotherapy prior to cytoreduction, use of a variety of different chemotherapy dosing strategies (e.g., weekly paclitaxel with or without weekly carboplatin), and the addition of antiangiogenic drugs (e.g., IV bevacizumab) into first-line treatment, followed by maintenance [7]. While efforts have shown maintenance treatment with poly (ADP-ribose) polymerase (PARP) inhibitors with or without bevacizumab plays a role in prolonging progression-free survival (PFS) in appropriately selected patients, it is not yet clear if this translates to improved overall survival; hence, outcomes remain poor.

The majority of patients with ovarian cancer will have intraperitoneal (IP) disease on progression [8,9,10]. Aggressive surgery (optimal debulking) significantly reduces the risk of IP recurrence (from 81% to 55%, *p* < 0.001), but does not eliminate it [9]. Similarly, in patients with optimally resected disease, the addition of IP chemotherapy to IV chemotherapy has been reported to improve outcomes compared to IV chemotherapy alone, but its use is limited [11], and treatment discontinuation predominantly relates to toxicity [2,12].

Ovarian cancer cells have a propensity to shed from the primary tumor into the peritoneal cavity, where they survive in ascites [12,13]. During the course of chemotherapy, and subsequent relapses, the proportion of stromal cells in the ascites gradually reduces and is replaced by chemoresistant cancer stem cells (CSCs) [14]. A defining characteristic of these CSCs is their capacity for self-renewal and differentiation [15,16], supporting their ability to regenerate a complex tumor population from very small numbers [17]. IP disease recurrence post-chemotherapy is thought to result in part from the growth of these chemoresistant CSCs and the formation of extraovarian peritoneal adhesions [13,14]. The addition of an effective front-line agent that specifically sensitizes CSCs to first-line chemotherapy has the potential to improve responses to platinum chemotherapy, leading to better clinical outcomes, longer overall survival, and a better quality of life for patients during treatment [18]. Novel therapies that can target these cells are therefore of great interest [12,13,19].

Early clinical trials with phenoxodiol (a first-generation benzopyran molecule) in combination with cisplatin or paclitaxel in advanced ovarian cancer demonstrated its ability to sensitize chemoresistant ovarian cancer cells to further chemotherapy [20,21,22]. However, these promising results were limited by poor bioavailability when phenoxodiol was administered IV. Compared to simple benzopyrans, third-generation benzopyrans have been shown to have a 1–2-log-fold increase in anticancer potency in preclinical experiments [23]. TRX-E-002-1 (Cantrixil; Kazia Therapeutics, Sydney, Australia, Licensed to Oasmia Pharmaceutical AB, March 2021) is the active enantiomer of a novel third-generation benzopyran molecule. Results from in vitro experiments using ovarian CSCs demonstrated that short exposure to high Cantrixil concentrations (2.45 μM) was optimal to prevent ovarian CSC recovery [24]. In addition, in vivo, IP Cantrixil monotherapy significantly reduced carcinomatosis in a murine ovarian cancer model [24]. When combined with cisplatin, administered IP in the same model, Cantrixil attenuated tumor kinetics and, consequently, delayed tumor recurrence post-paclitaxel therapy. Given the propensity of ovarian cancer cells to remain within the peritoneal cavity [12,13], patterns of relapse that point primarily to the peritoneum, and the accumulation of chemoresistant CSCs, IP drug delivery is an attractive approach for recurrent ovarian cancer [25]. The decision to develop Cantrixil for use as an IP infusion was therefore based on a desire to overcome IV bioavailability limitations and maximize local drug–cancer cell contact.

Cantrixil offers a potential means of improving long-term survival by reducing rates of recurrence through the elimination of ovarian CSCs and retarding tumor growth through its activity against non-CSC bulk cancer cells. This open-label, Phase I clinical study (NCT02903771) evaluated the safety and feasibility of administering weekly IP Cantrixil in heavily pretreated patients with persistent or recurrent epithelial ovarian cancer, fallopian tube cancer, or primary peritoneal cancer. To our knowledge, it is the first in-human study to demonstrate the potential for prolonged survival in advanced ovarian cancer by directly targeting chemoresistant IP CSCs.

## 2. Materials and Methods

### 2.1. Study Design

This progressive design, multicenter study was conducted in two discrete parts: Part A was a dose escalation phase to determine the maximum tolerated dose (MTD) of Cantrixil to be administered in Part B, an expansion cohort phase (Figure 1). The study was prospectively registered, conducted in accordance with the ethical principles established in the Declaration of Helsinki, and all participants provided written informed consent. Data were collected between 5 December 2016 and 24 March 2020.

### 2.2. Patient Population

Female patients, ≥18 years, with recurrent or persistent epithelial ovarian, primary peritoneal, or fallopian tube cancer (of any histological subtype and grade) were eligible provided they had completed at least two prior regimens and had investigator-determined recurrent platinum-resistant (Pt-R) disease, platinum-refractory (Pt-Rf) disease, or documented intolerance to platinum therapy (Pt-S). Patients were ineligible based on rising cancer antigen-125 (CA-125) levels alone; clinical symptoms or radiological tumor measurements that supported disease recurrence/progression were required. Other eligibility criteria included physical suitability for treatment (Eastern Cooperative Oncology Group (ECOG) performance status 0–2), willingness to undergo insertion of a port/catheter for IP access, and acceptable hepatic, renal, and bone marrow function. Patients were excluded if they had received chemotherapy, biologic therapy, immunotherapy, or radiotherapy within 4 weeks (6 weeks for bevacizumab, nitrosoureas, or mitomycin C) of study entry, major surgery within 4 weeks of screening, had evidence of uncontrolled or severe systemic disease, were at high risk of bowel perforation or obstruction, or were immunocompromised. Patients enrolled into Part A were not permitted to continue to Part B.

### 2.3. Treatments Administered

Part A (dose escalation), Cycle 1 was commenced with Cantrixil 0.24 mg/kg, administered as a once-weekly IP dose, over a 3-week cycle (3 doses in total). Successive single-patient cohorts were treated with increasing doses of Cantrixil (Appendix A) until an adverse event meeting the definition of a dose-limiting toxicity was observed. At this point, the study reverted to a 3-plus-3, rules-based dose escalation design. The MTD was defined as the highest dose level at which no more than one of six patients in a cohort experienced a dose-limiting toxicity. Patients continued on weekly IP Cantrixil monotherapy for two cycles. In Cycles 3–8 patients continued with the same weekly dose of IP Cantrixil, administered in combination with an IV chemotherapy protocol selected for the patient by the treating physician from a pre-specified list of regimens (Appendix A). Chemotherapy was administered 24 h after Cantrixil in order to mitigate any adverse drug–drug interactions. Part B (dose expansion) commenced after the MTD had been determined; patients received the same treatment as described above.

### 2.4. Assessments and Evaluations

Patients underwent weekly assessments prior to Cantrixil administration (physical examination, vital signs, ECOG performance status, safety laboratory assessments, and CA-125 levels). Biomarker samples were collected at screening, at the end of Cycle 2, and at the end of therapy during Part A of the study only. Additional blood samples were collected before and after each Cantrixil administration for pharmacokinetic analyses. Tumor assessment was conducted via radiological imaging at screening, and then every 6 weeks (i.e., at the end of each 2-cycle period), and/or at any time when progression of disease was suspected. Adverse events were evaluated according to the National Cancer Institute Common Terminology Criteria for Adverse Events, version 4.03.

### 2.5. Study Endpoints

The primary objective of Part A was to determine Cantrixil’s MTD. The primary objectives of Parts A and B were to evaluate the safety and characterize the pharmacokinetic profile of Cantrixil monotherapy and in combination with standard chemotherapy. Secondary endpoints included progression-free survival (PFS; time from treatment commencement until objective disease progression, as defined by Response Evaluation Criteria in Solid Tumors (RECIST), version 1.1), time to paracentesis, and volume of malignant ascites, and evaluation of the clinical activity of Cantrixil based on investigator-assessed objective response rate (ORR) and disease control rate (DCR). Evaluation of the changes in number and clonogenicity of circulating epithelial tumor cells (CETCs) and expression of stem cell markers CD44 and ALDH were exploratory endpoints, conducted in patients enrolled in Part A of the study.

### 2.6. Exploratory Analysis of Stem Cell Markers

Exploratory analysis of stem cells was conducted in patients enrolled into Part A of the study only. Samples for analysis were obtained at baseline, at the end of Cycle 2 (week 6), and at the end of therapy. Enumeration of CETCs in peripheral blood and malignant ascites (where present) was assayed using the Maintrac^®^ (DMB Diagnostics, Berlin, Germany) CETC Count method. CETCs were tagged using a fluorochrome-labelled antibody against the surface epithelial antigen molecule (EpCam) and image analysis conducted to ascertain CETC numbers in relation to blood volume over the course of treatment. Clonogenicity of CETCs from the same samples was measured using the Maintrac^®^ Tumor Sphere Units assay. Expression of stem cell markers aldehyde dehydrogenase (ALDH) and CD44 in the isolated colonies was measured using fluorescein isothiocyanate-labelled antibodies and scanning fluorescent microscopy techniques.

### 2.7. Statistical Considerations

Demographics, baseline characteristics, safety, and pharmacokinetic data were all summarized descriptively. Response endpoints were evaluated using the efficacy population, defined as all patients who had received at least one dose of Cantrixil and who had at least one post-baseline tumor assessment available. Given that the primary endpoint was to determine the MTD, an appropriate sample size was not statistically determined. Kaplan–Meier estimates were used to obtain median PFS and the corresponding 2-sided 95% confidence intervals (CIs). The best overall response was derived using published definitions [26]. The database was locked on 24 August 2020, and statistical analyses performed using SAS software, version 9.2 or higher (SAS Institute Inc, Cary, NC, USA).

## 3. Results

### 3.1. Patient Population and Baseline Characteristics

Of the 32 patients enrolled, 25 patients received at least one dose of Cantrixil (Figure 2). Demographics and baseline characteristics for the patients in Part A, for those patients receiving the Cantrixil MTD, and for all patients (Parts A and B combined) are shown in Table 1.

### 3.2. Part A: Determination of MTD

During Part A of the study, 11 patients were enrolled into 6 dose-level cohorts (Figure 3). Following the successive single-patient cohort dose escalation design, no dose-limiting toxicities had been experienced by the first patient enrolled at the 10.0 mg/kg dose level, and the dose level was increased to 20.0 mg/kg. This patient experienced grade 3 ileus commencing 4 days after the second dose of Cantrixil and lasting 2 days, and experienced grade 3 abdominal pain after the third dose; the treatment was discontinued. Subsequent to this, the 10.0 mg/kg dose level was expanded to include a second patient, who experienced a dose-limiting toxicity of grade 3 ileus and grade 3 abdominal pain after receiving the second Cantrixil dose. Two further patients were enrolled into the 5.0 mg/kg dose level—one of whom experienced grade 3 intestinal obstruction and grade 3 abdominal pain, which was symptomatic of bowel obstruction. Based on the dose-limiting toxicity of the grade 3 ileus, at the 20 mg/kg (1 patient) and 10 mg/kg doses (1 patient), and the observed safety signals of bowel obstruction and abdominal pain, the Cantrixil MTD was established as 5.0 mg/kg administered via IP infusion once weekly.

### 3.3. Pharmacokinetics

The pharmacokinetic profile of IP-administered Cantrixil was multiexponential, and characterized by a rapid increase in systemic concentration, followed by a slower elimination phase. Peak plasma concentrations were observed 0.3 to 6 h after dose administration, and progressively declined such that they were <10% of maximal concentrations by 24 h post-dosing. There was minimal Cantrixil accumulation, and co-administration of chemotherapy had no impact on its pharmacokinetic profile. Graphical representation of pharmacokinetic parameters as a function of dose indicates no clear dose-related trends (Appendix A). Substantial interindividual variability in the pharmacokinetic profile of Cantrixil was observed, particularly in the dose expansion cohort. Exploration of the relationship with patient factors found no clear associations between dose-normalized area under the curve (AUC)_last_, AUC_inf_, and C_max_ values and age, height, and tumor type. A positive relationship between weight (and, by extension, body mass index) and dose-normalized (to 1 mg/kg)) AUC_last_, AUC_inf_ and C_max_ values was observed, but further graphical analysis indicated no notable trends (Appendix A).

### 3.4. Safety

The most common treatment-related adverse events of any grade at any Cantrixil dose were abdominal pain (*n* = 12, 48%), vomiting (*n* = 10, 40%), fatigue (*n* = 9, 36%), and nausea (*n* = 7, 28%) (Appendix A). Treatment-related adverse events that occurred during monotherapy and combination therapy are summarized in Table 2. The most frequently occurring grade 3 treatment-related adverse events were abdominal pain (*n* = 5, 20%), vomiting (*n* = 2, 8%), and ileus (*n* = 2, 8%) during monotherapy, and neutropenia (*n* = 2, 8%) during combination therapy (Table 2). The two patients who experienced grade 3 ileus during Part A of the study were among three patients who had a body mass index of >35 kg/m^2^ (obese class 2) at enrolment. Overall, seven (28%) patients discontinued Cantrixil treatment due to adverse events; all discontinuations occurred during monotherapy (three patients were enrolled in Part A; one each with a Cantrixil dose of 0.24 mg/kg, 10 mg/kg, and 20 mg/kg). Three (12%) patients withdrew from the study due to adverse events. There were no treatment-related deaths. Two patients died during the study because of disease progression; both patients were enrolled in Part B of the study, and had received Cantrixil at the MTD of 5.0 mg/kg. There were no clinically meaningful changes in mean clinical laboratory measurements for hematology, serum biochemistry, or urinalysis, and the only notable observations during the study were reports of abnormal abdominal and gastrointestinal symptoms, which is consistent with the adverse events reported.

### 3.5. Clinical Findings: Antitumor Activity

Of the 25 patients treated in Parts A and Part B of the study, 9 were not evaluable for efficacy—7 of whom had no available post-baseline efficacy assessment, and 2 (who had both received less than one full 3-week cycle of Cantrixil) who had post-baseline assessments conducted only as part of an end-of-study visit subsequent to discontinuation of treatment due to an adverse event. In the efficacy-evaluable population (*n* = 16 (64%); any Cantrixil dose, Figure 4), the best overall response after monotherapy was stable disease (SD; nine patients), and after combination therapy one patient had a confirmed complete response (CR), two patients had partial responses (PR), six had SD, and seven had progressive disease (PD). The confirmed overall response rate (ORR) in the efficacy-evaluable population was 18.8% and, based on RECIST Version 1.1 criteria, the median PFS was 13.1 weeks (95% CI: 5.56, ∞; Figure 5). Post-hoc subgroup analysis of efficacy in patients (*n* = 11) confirmed to have either Pt-Rf disease or Pt-R disease showed an improved median PFS of 19.4 weeks (95% CI: 5.84, ∞; Figure 6a), which was further improved to 23.8 weeks (95% CI: 5.00, ∞; Figure 6b) in the cohort of three patients with Pt-Rf disease.

### 3.6. Translational Research

During the study, four patients (all in Part B, receiving the Cantrixil MTD of 5.0 mg/kg) required paracentesis. No trends were observed in either the normalized time between these events or the volume of fluid drained. Three of the patients undergoing paracentesis had a best overall response of SD, while the fourth experienced PD. There was a high degree of interindividual variation in serum CA-125 concentrations (range at screening: 5–5360 U/mL). In the population overall, there was an initial increase in median CA-125 concentrations between screening and the end of the Cantrixil monotherapy (weeks 1–6), followed by subsequent reduction during combination therapy (weeks 7–24). This general trend of a reduction in median CA-125 concentrations during combination therapy was also observed in subgroup analyses of patients with Pt-S disease and Pt-R disease, but not in patients with Pt-Rf disease.

Preliminary analysis of stem cell markers, in this limited dataset, demonstrated a 3-fold reduction in spheroid cells and a 12-fold reduction in ALDH between screening and the end of Cycle 2 (Figure 7). Of note, of the three patients who responded to Cantrixil, two had no spheroid cells present at screening; the third patient, who had Pt-Rf disease and achieved an overall PR to Cantrixil 5.0 mg/kg, showed complete elimination of spheroid cells (400 at screening to 0 at end of treatment). Spheroid cells were eliminated in two other patients: 600→0 in a patient with Pt-S disease and a best overall response of SD to Cantrixil 5.0 mg/kg, and 73→0 in a patient with Pt-Rf disease and a best overall response of SD to Cantrixil 0.24 mg/kg. There were no notable trends in the analyses of number and clonogenicity of CETCs, or in the expression of CD44 (Appendix A).

## 4. Discussion

This first in-human Phase I clinical study has established IP-administered Cantrixil to have an MTD of 5.0 mg/kg, with the dose-limiting toxicity of grade 3 ileus. At this MTD, Cantrixil was well tolerated, and demonstrated a multiexponential pharmacokinetic profile, characterized by a rapid increase in systemic concentrations after the end of the infusion, followed by a rapid distributional phase and a slower elimination phase, which was not impacted by concomitant IV chemotherapy.

In disseminated ovarian cancer, IP infusion offers the potential for superior delivery of small molecule therapies to avascular tumors in patients with ascites and small, neovascularized tumors seeding the peritoneal cavity [27]. In preclinical pharmacokinetic experiments, IP Cantrixil administration showed tissue (pancreas, ovaries, large intestine, skin, kidneys, liver, stomach, and adrenal glands) levels to be higher than the plasma concentration [28]. A key consideration with IP administration, however, is how to ensure even distribution of the drug within the peritoneal cavity. Cantrixil (100 mL) was infused first under gravity, followed by infusion of up to 1 L of warmed saline to facilitate distribution. Administration procedures followed previously established guidelines for IP chemotherapy, and included careful monitoring of the needle position and repositioning of the patient every 15 min for 1 h post-infusion [29]. Incorporation of Cantrixil into earlier lines of treatment, prior to the development of dense adhesions, could be employed as another strategy to promote even drug distribution.

To date, observed improvements with upfront treatment of advanced ovarian cancer with IP chemotherapy [30,31] have been offset against a less favorable toxicity profile [31,32,33], which was frequently associated with significant abdominal pain [34]. Overall, 16 (64%) patients experienced at least one grade 3 gastrointestinal adverse event; of these, 3 patients consequently withdrew from the study—one due to the dose-limiting toxicity of ileus/abdominal pain at a dose of 20 mg/kg, one due to bowel obstruction at a dose of 0.24 mg/kg, and one due to a small intestinal obstruction at a dose of 5.0 mg/kg. While the MTD of 5.0 mg/kg was established on the basis of the gastrointestinal findings, it is difficult to delineate whether the most frequently reported adverse events during Cantrixil monotherapy (vomiting, nausea, abdominal pain, and bowel obstruction) arose from the Cantrixil itself, or were a consequence of the IP port. Further exploration of the cause of the grade 3 abdominal pain is required in order to better define the toxicity profile of Cantrixil.

Comparison of dose-normalized data indicates that Cantrixil does not display any clear lack of dose proportionality over the doses tested. Data from our pharmacokinetic analyses are suggestive that body mass index may not be of relevance, and that flat dosing could be an appropriate option to investigate in future trials. Such a strategy could have implications for improving the toxicity profile of Cantrixil, given that both cases of grade 3 ileus were observed in Part A of the study in patients with a body mass index > 38 kg/m^2^ who had received doses above the MTD.

The prognosis of patients with advanced ovarian cancer is poor. Most patients succumb to disease progression [6], and will die of platinum-resistant disease, primarily due to the accumulation of chemoresistant CSCs in the peritoneal cavity [12,13,14]. The ability to target chemoresistant CSCs has been a major barrier to the development of novel therapies for patients with advanced disease. Our results show promising antitumor activity in monotherapy (SD achieved in 56% of patients), and when IP Cantrixil is combined with chemotherapy (ORR 19%, DCR 56%, and median PFS 13.1 weeks). This disease response compares favorably to a figure of 10% for historical controls and is largely consistent with other recent research investigating IP treatment regimens in platinum-resistant ovarian cancer.

In a Phase III trial, IP cisplatin (40 mg/m^2^) plus bevacizumab (300 mg), every 2 weeks for 6 weeks, was superior to the same dose of IP cisplatin alone (ORR 90.32 vs. 59.26%, *p* < 0.05), and resulted in significantly reduced VEGF and CA-125 levels in ascites fluid [35]. Early data suggestive of a role for IP bevacizumab in the palliation of malignant ascites in recurrent ovarian cancer [36,37] are supported by clinical experience demonstrating improved ascites with IP-administered bevacizumab (5 mg/kg) in 6/7 (85.6%) patients [38]. The Phase II REZOLVE study has demonstrated that IP bevacizumab (5 mg/kg), administered following paracentesis, increased the median time of paracentesis-free intervals by 4.3-fold compared to the time between paracenteses before entry into the study [39]. In Pt-R patients who had received IP bevacizumab plus IV cisplatin for the treatment of malignant ascites, clinical response correlated with IP immune activation [40], and compared to non-responders the responding patients had higher levels of circulating effector CD4 T cells and lower IL-10 cytokine levels [41]. Exploratory analyses for potential predictive biomarkers to bevacizumab from the REZOLVE trial, including VEGF pathway factors in blood and ascites, are yet to be reported [39]. Preclinical findings suggesting that bortezomib may enhance tumor platinum uptake by targeting the cell membrane copper transport receptor [42] have been confirmed in a Phase I Gynecologic Oncology Group trial, which showed positive results (ORR 19%) after administration of IP bortezomib in combination with IP carboplatin in recurrent, heavily pretreated disease [43]. Despite initial interest in the potential for the combined use of PARP inhibitors and IP fluoropyrimidines [44], results of a recently completed Phase I study evaluating the combination of IV veliparib and IP floxuridine (NCT01749397) have not yet been reported [45].

Our findings have demonstrated a potential for prolonged survival in a population of patients with very poor prognosis. A plethora of studies with emerging targeted therapies have reported promising data in heavily pretreated late-stage ovarian cancer, but relatively few have included patients with confirmed Pt-Rf disease. These patients have few therapy options, but due to their poor prognosis they are under-represented, accounting for only around 10% of the total population recruited in many clinical studies [46,47], including our own. Several early-phase trials with targeted therapies have reported lower ORR in subgroups of patients with Pt-Rf disease than in the overall study population [46,48,49], indicating that response may be related to platinum sensitivity status—this is the reverse of our findings. Post-hoc exploratory analyses showed an enhanced rate of response amongst patients with Pt-Rf disease (median PFS 23.8 weeks versus 13.1 weeks for the primary analysis population), suggesting that the degree of platinum resistance could potentially correlate with superior survival outcomes with Cantrixil.

The preliminary analysis of stem cell markers suggests that inducing ovarian CSC death and sensitizing cancer cells to standard chemotherapy could be key aspects of the mechanism by which weekly IP Cantrixil has the potential to prolong survival in patients with heavily pretreated advanced ovarian cancer. The presence of CSCs in the ascites plays an important role in disease recurrence (via self-renewal and differentiation) and in the development of treatment resistance by entering a quiescent state [50,51]. Initial research investigating CSC targeting as a means of overcoming disease recurrence focused on therapies directed at stemness markers, including bivatuzumab and imatinib, but has been hampered by safety issues because they shared stemness factors that are associated with normal stem cells [52]. More recent preliminary studies with other approaches, including epigenetic therapies (e.g., guadecitabine) and signaling pathway inhibitors (e.g. Hedgehog, Hippo, Wnt, Jak2/STAT3, PI3K/PTEN/AKT, NF-κB, and Notch3) have begun to report, with varying results [52]. A clear direction that has come from this research is the potential dual role for targeted therapies in chemoresistant patients, whereby they not only influence CSC development, but may also play a role in restoring chemosensitization. On entering the combination therapy phase of the trial, women were eligible to receive standard of care chemotherapy, as deemed suitable by their treating clinician. While this provided for the best possible outcome with regards to individual patient choice in this early phase study, it precluded the ability to distinguish the true contribution of IP Cantrixil from that of the IV chemotherapy. However, to date, the patient with Pt-R disease who achieved a complete response to Cantrixil 2.5 mg/kg during the study has not relapsed. This extended response (currently > 36 months post study completion) suggests that targeting chemoresistant CSCs with IP Cantrixil has influenced her long-term survival.

While the molecular target of Cantrixil is yet to be confirmed, it has been identified as having potent cytotoxicity against chemoresistant CD44+/MyD88+ ovarian CSC clones and chemosensitive CD44-/MyD88- ovarian cancer cell lines [28]. Preclinical studies suggest that Cantrixil induces apoptotic cell death by both caspase-dependent and caspase-independent apoptosis, the mechanism of which may involve tumor-associated NADH oxidase (ENOX2) and disruptions to transmembrane electron-transport-mediated energy production [17,28]. Observations that Cantrixil upregulates proapoptotic factors (e.g., Bax, bid, and X-chromosome-linked inhibitor of apoptosis protein (XIAP)) plus phospho-c-Jun suggest that the mechanism invokes the activation of oxidative stress pathways associated with mitochondrial depolarization [23]. Data from in vitro experiments demonstrate more than three out of five platinum-resistant ovarian cancer cell lines to be susceptible to Cantrixil [24]. Adding Cantrixil to cultured spheroids of human ovarian CSCs destroyed the spheroid structures, confirming the ability of the drug to penetrate three-dimensional structures and to kill the cancer cells [24]. Although exploratory analyses of CSC biomarkers were not successful in the current study, in a limited dataset we demonstrated complete elimination of spheroid cells. High ALDH enzyme activity within ovarian CSCs has been implicated in drug resistance [53] and poor overall survival. Although preliminary, our observed 12-fold reduction in ALDH supports prior preclinical data in which downregulation of ALDH sensitized ovarian CSCs to chemotherapy [54]. Identifying predictive biomarkers that could be used to select patients with a greater likelihood of clinical benefit remains a high priority.

## 5. Conclusions

This Phase 1 study has established the tolerability of IP-administered Cantrixil, with an MTD of 5.0 mg/kg. It has demonstrated that IP administration of Cantrixil is feasible, efficacious at targeting chemoresistant CSCs, and results in a reasonable prolongation of survival in a heavily pretreated population. The results are compelling, but cautious interpretation is warranted until they can be validated through further functional studies and larger clinical trials in earlier lines of treatment. Future functional studies should focus on further delineating its mechanism of action, and a more rigorous assessment of its effects on CSC biomarkers. Potential avenues of clinical investigation include evaluation of the role of Cantrixil in combination with neoadjuvant/adjuvant chemotherapy to help eradicate CSC clones in patients with advanced disease undergoing interval debulking. However, findings to date point towards establishing Cantrixil as a chemosensitizer and targeted inducer of ovarian CSC cell death, suggesting that its utility could be greatest as an up-front therapy to eliminate CSCs in chemotherapy-naïve patients.

## Figures and Tables

**Figure 1 cancers-13-03196-f001:**
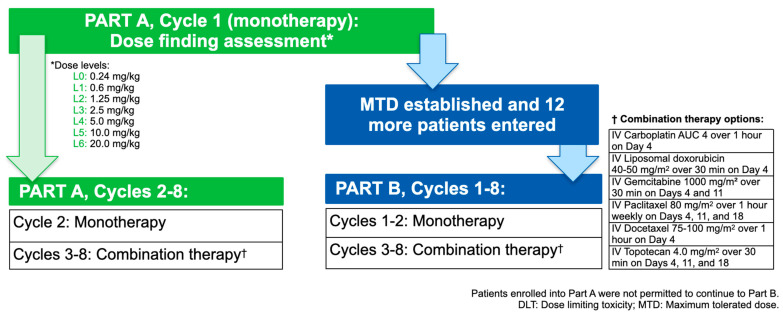
Schematic of the study design.

**Figure 2 cancers-13-03196-f002:**
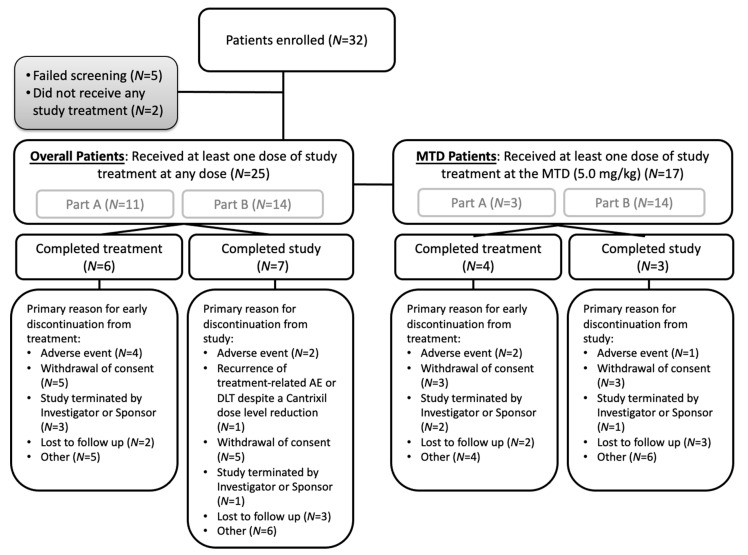
Patient disposition, with separate details of the overall patient population and the patients included in the MTD population. AE: adverse event; DLT: dose-limiting toxicity; MTD: maximum tolerated dose. Patients were defined as enrolled when signing informed consent. Figure includes patients from Parts A and B receiving at least 1 dose of Cantrixil (Overall Patients) and at least one dose of Cantrixil at the MTD of 5.0 mg/kg (MTD Patients).

**Figure 3 cancers-13-03196-f003:**
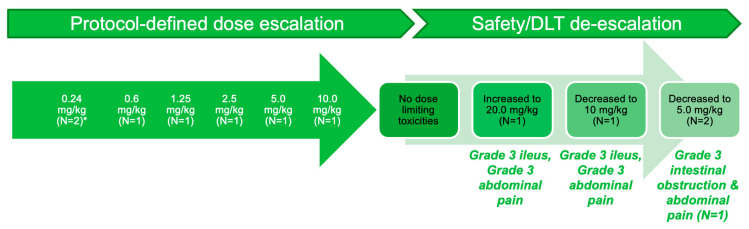
Study Part A: Results of dose-finding assessment. Visual depiction of the number of patients enrolled into pre-specified dose cohorts, and the toxicities upon which the maximum tolerated dose was established. * Two patients were enrolled into the 0.24 mg/kg dose cohort—the first patient experienced intestinal obstruction commencing 5 days after receiving her first dose of Cantrixil; this event was not deemed to be study drug related, but the patient was subsequently withdrawn from the study, and a second patient was enrolled at the same dose level. DLT: dose-limiting toxicity; MTD: maximum tolerated dose.

**Figure 4 cancers-13-03196-f004:**
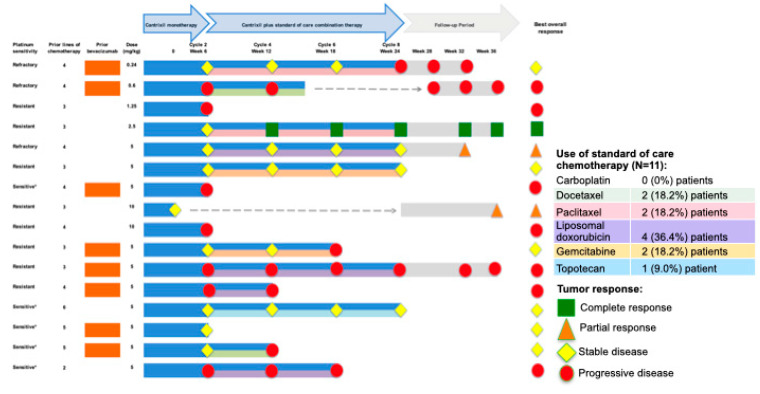
Swimmer plot showing duration of therapy, combination therapy regime, and disease response based on RECIST Version 1.1 criteria for each patient. * Patients who were unable to receive further platinum-based chemotherapy due to documented intolerance.

**Figure 5 cancers-13-03196-f005:**
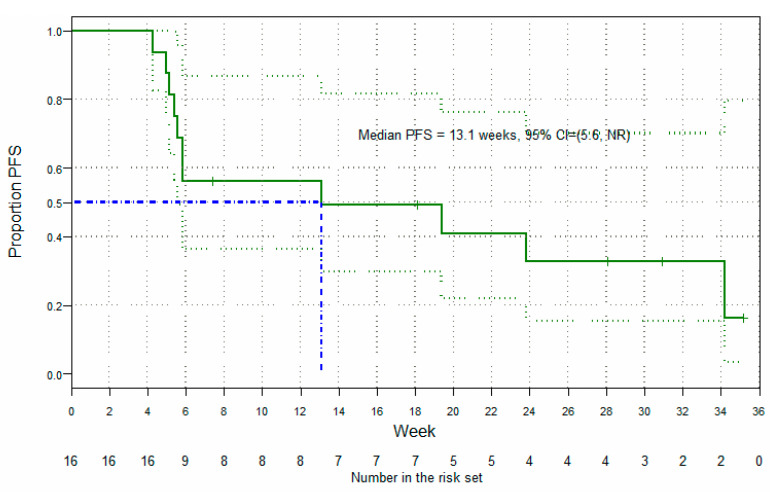
Secondary endpoint: Kaplan–Meier survival curve in the efficacy-evaluable population.

**Figure 6 cancers-13-03196-f006:**
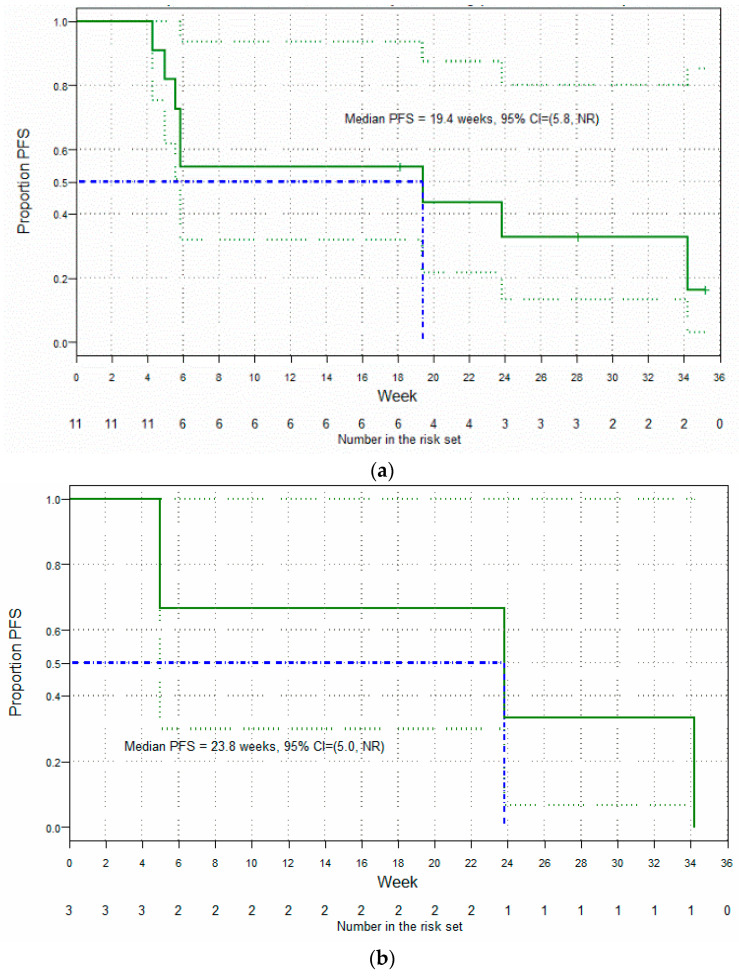
Post-hoc analysis of the antitumor activity of Cantrixil: (**a**) Kaplan–Meier survival curve in the efficacy-evaluable population, minus patients with platinum-sensitive disease (Patients who were unable to receive further platinum-based chemotherapy due to documented intolerance); (**b**) Kaplan–Meier survival curve in patients with platinum-refractory disease.

**Figure 7 cancers-13-03196-f007:**
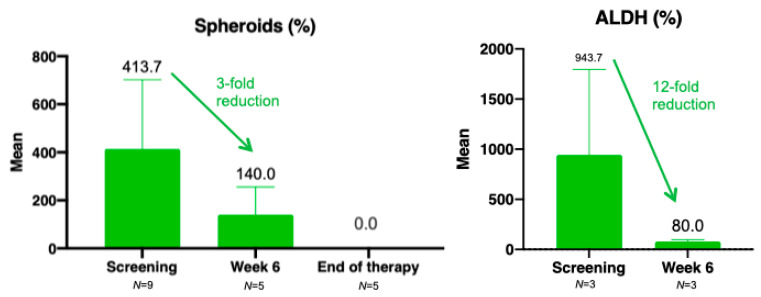
Exploratory analysis of stem cell markers (population: Part A only). ALDH: aldehyde dehydrogenase. Data are plotted as the mean (standard error of the mean). Week 6 = end of Cycle 2.

**Table 1 cancers-13-03196-t001:** Patient characteristics at baseline.

Characteristic	Part A	Patients Receiving MTD ^a^	All Patients
	*n* = 11	*n* = 17	*n* = 25
Age, median (range), y	66 (42–79)	61 (46–79)	62 (42–79)
BMI, median (range), kg/m2	28.2 (20.1–38.3)	29.2 (18.8–36.6)	28.2 (18.8–38.3)
Time since diagnosis, median (range), m	52.7 (17.2–87.8)	62.3 (14.0–255.8)	48.5 (14.0–255.8)
ECOG performance status, *n* (%)			
0	8 (72.7)	6 (35.3)	12 (48.0)
1	3 (27.3)	11 (64.7)	13 (52.0)
Tumor stage at study entry, *n* (%)			
III	4 (36.4)	6 (35.3)	10 (40.0)
IV	7 (63.6)	11 (64.7)	15 (60.0)
Tumor type, *n* (%)			
Epithelial ovarian cancer	9 (81.8)	11 (64.7)	18 (72.0)
Fallopian tube cancer	1 (9.1)	2 (11.8)	3 (12.0)
Primary peritoneal cancer	1 (9.1)	4 (23.5)	4 (16.0)
Histological subtype, *n* (%)			
Serous Carcinoma	8 (72.7)	11 (64.7)	17 (68.0)
Endometrioid Carcinoma	1 (9.1)	1 (5.9)	1 (4.0)
Clear Cell Carcinoma	0 (0.0)	1 (5.9)	1 (4.0)
Other	2 (18.2)	4 (23.5)	6 (24.0)
Prior lines of therapy, median (range)	4 (3–6)	5 (2–11)	5 (2–11)
Prior chemotherapy, *n* (%)			
Platinum	11 (100.0)	17 (100.0)	25 (100.0)
Taxane	11 (100.0)	17 (100.0)	25 (100.0)
Anthracycline	11 (100.0)	12 (70.6)	19 (76.0)
Gemcitabine	7 (63.6)	11 (64.7)	15 (60.0)
Melphalan	0 (0.0)	3 (17.6)	3 (12.0)
Topotecan	0 (0.0)	3 (17.6)	3 (12.0)
Cyclophosphamide	0 (0.0)	1 (5.9)	1 (4.0)
Prior bevacizumab, *n* (%)	4 (36.4)	10 (58.8)	13 (52.0)
Platinum status, *n* (%)			
Refractory (Pt-Rf)	3 (27.3)	1 (5.9)	3 (12.0)
Resistant (Pt-R)	7 (63.6)	11 (64.7)	17 (68.0)
Sensitive, but intolerant ^b^ (Pt-S)	1 (9.1)	5 (29.4)	5 (20.0)
BRCA status, *n* (%)			
BRCA1 mutation	1 (9.1)	3 (17.6)	3 (12.0)
BRCA2 mutation	0 (0.0)	1 (5.9)	1 (4.0)
Negative	7 (63.6)	12 (70.6)	17 (68.0)
Unknown	3 (27.3)	1 (5.9)	4 (16.0)

^a^: Patients from either Part A or Part B who received Cantrixil 5 mg/kg. ^b^: Patients who were unable to receive further platinum-based chemotherapy due to documented intolerance. BMI: body mass index; BRCA: breast cancer susceptibility gene; ECOG: Eastern Cooperative Oncology Group; MTD: maximum tolerated dose.

**Table 2 cancers-13-03196-t002:** Treatment-related adverse events during monotherapy and combination therapy occurring in ≥2 patients.

Number (%) of Patients with Adverse Event, MedDRA Preferred Term	Part A		Patients Receiving MTD ^a^		All Patients	
	*n* = 11		*n* = 17		*n* = 25	
Monotherapy	Grade 1–2	Grade 3 ^b^	Grade 1–2	Grade 3 ^b^	Grade 1–2	Grade 3 ^b^
Any adverse event	4 (36.4)	5 (45.5)	5 (29.4)	7 (41.2)	8 (32.0)	10 (40.0)
Abdominal pain	4 (36.4)	3 (27.3)	1 (5.9)	4 (23.5)	5 (20.0)	5 (20.0)
Vomiting	4 (36.4)	1 (9.1)	6 (35.3)	1 (5.9)	8 (32.0)	2 (8.0)
Nausea	4 (36.4)	1 (9.1)	4 (23.5)	0 (0.0)	6 (24.0)	1 (4.0)
Diarrhea	3 (27.3)	0 (0.0)	2 (11.8)	0 (0.0)	4 (16.0)	0 (0.0)
Abdominal distension	2 (18.2)	0 (0.0)	3 (17.6)	0 (0.0)	4 (16.0)	0 (0.0)
Abdominal discomfort	1 (9.1)	0 (0.0)	1 (5.9)	0 (0.0)	2 (8.0)	0 (0.0)
Ileus	0 (0.0)	2 (18.2)	0 (0.0)	0 (0.0)	0 (0.0)	2 (8.0)
Fatigue	5 (45.5)	0 (0.0)	3 (17.6)	0 (0.0)	8 (32.0)	0 (0.0)
**Combination therapy**						
Any adverse event	2 (18.2)	1 (9.1)	2(11.8)	2(11.8)	3 (12.0)	2 (8.0)
Abdominal pain	2 (18.2)	0 (0.0)	2 (11.8)	0 (0.0)	3 (12.0)	0 (0.0)
Neutropenia	0 (0.0)	1 (9.1)	0 (0.0)	1 (5.9)	0 (0.0)	2 (8.0)

^a^: Patients from either Part A or Part B who received Cantrixil 5 mg/kg; ^b^: There were no grade 4 or 5 adverse events. MedDRA: Medical Dictionary for Regulatory Activities; MTD: maximum tolerated dose.

## Data Availability

The data that support the findings of this study are available from the corresponding author upon reasonable request.

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
