# Peer review of "Maximum Tolerated Dose and Anti-Tumor Activity of Intraperitoneal Cantrixil (TRX-E-002-1) in Patients with Persistent or Recurrent Ovarian Cancer, Fallopian Tube Cancer, or Primary Peritoneal Cancer: Phase I Study Results"

_cancers, 2021, doi:10.3390/cancers13133196_

Round 1

Reviewer 1 Report

The authors present in this manuscript the data from a phase I study evaluating the safety and feasibility of administering weekly ip Cantrixil in heavily pre-treated patients with persistent or recurrent ovarian cancer. The significance of the results is obvious, but the manuscript needs some revision before it can be recommended for publication.

Simple summary

I will suggest to simply the statement “Cantrixil is a novel third-generation benzopyran 21 molecule previously identified as having potent cytotoxicity against chemo-resistant 22 CD44+/MyD88+ ovarian CSC clones and chemo-sensitive CD44-/MyD88- ovarian cancer cell lines”  In order to keep the readers interest.

Abstract

The target as well as mechanisms of action for Cantrixil should be mentioned.

Introduction

  • This paragraph is clear and easy to follow. However, a more detailed description of the biological rational for selecting Cantrixil as well as the drugs mechanisms of action are needed.

Materials and Methods

  • The information should be presented in a more systematic way. I will suggest to structure the information into separate paragraphs, like study design, ethics, patient population, treatment, study assessments, pharmacokinetic assessments, translational research (the different methods) and statistical analysis.

Results

  • Also this part is difficult to assess and also for this I will suggest a different structure; patient population, dose limiting toxicity, pharmacokinetics, clinical activity and translational research.
  • The figure legends must be more descriptive

Discussion

  • The discussion is biologically focused, and gives a better understanding of the treatment approach. However I will suggest that the authors select 3 or 4 major findings before they discuss and interpreter them separately and in context of the literature published. First at the end the focus should be both on the biological relevance and the clinical consequences of the findings.

Author Response

Please see the attached word file, a summary of our responses to this reviewers comments is also provided below.

Simple summary

This sentence of the summary has been revised to read as follows:

“Cantrixil is a novel third-generation benzopyran molecule, with previously identified as having potent cytotoxicity against chemo-resistant CD44+/MyD88+ ovarian CSC clones cancer stem cells and chemo-sensitive CD44-/MyD88- ovarian cancer cell lines.”

Abstract

As explained in the discussion of the paper, the molecular target and mechanism of action of Cantrixil are yet to be confirmed and further study in this area is required. The last sentence of the abstract provides this information. However, to more specifically address this, we have added an additional explanatory sentence into the abstract.

“The molecular target and mechanism of action of Cantrixil are yet to be confirmed. Preliminary analysis of stem cell markers suggests that IP Cantrixil might induce ovarian cancer stem cell death and sensitize cells to standard chemotherapy; warranting further evaluation.”

Introduction 

Given the preliminary nature of this first-in-human study we had included the rationale for selecting Cantrixil and the rationale for IP administration in the discussion of the paper (see highlighted paragraph). However, it fits equally well in the introduction and, in response to this reviewer, we have moved it to the introduction.

As explained above, the molecular target and mechanism of Cantrixil have not yet been fully determined. Given this and the need for further evaluation, we feel that commentary on this would be best placed in the discussion section of the manuscript and have therefore made no change to the introduction.  

Materials and Methods 

The various paragraphs of the manuscript are already set out in the order suggested by the reviewer. For completeness, we have included sub-headings to better guide the reader, and defer to the Editor to determine if these are a suitable addition.

Results

As with the methods section, sub-headings have now been included to better guide the reader through the data presented.

Figure legends have been updated to be more explanatory, where possible.

Discussion

The discussion has been re-organised, and reworded in various places, to address this concern. It is now organized as follows:

  • MTD, PK and safety findings and discussion of these
  • Clinical activity and discussion of this
  • Translational research and discussion of this
  • Biological relevance and implications for further research

Reviewer 2 Report

The manuscript “Phase I Study of Intra-peritoneal Cantrixil (TRX-E-002-1) in Patients with Persistent or Recurrent Ovarian Cancer, Fallopian Tube Cancer or Primary Peritoneal Cancer” by Jermaine I. Coward et al. describes an interesting study on the use of Cantrixil  to improve the disease control rate and the progression free survival in advanced ovarian cancer.

-The results of the study are well presented but there are some comments:

-The authors should indicate more clearly the aim of the study in the summary and in the abstract of the manuscript.

- The title of the manuscript should be more specific reporting some words referred to the aim of the study

-The authors give great evidence to the concept of the drug resistance in tumor due to the presence of a cancer stem cell (CSC) subpopulation not responder to cytotoxic agents and potentially targetable with Cantrixil. The results showed in figure 7 are very interesting and give force to such thesis but the analysis was done only on three responder patients. The authors should comment in the results and in the discussion section of the manuscript the other possible causes of drug resistance in not responder or partially responder patients to Cantrixil.

Author Response

Please see the attached word file, a summary of our responses to this reviewers comments is also provided below.

The authors should indicate more clearly the aim of the study in the summary and in the abstract of the manuscript.

We have revised the wording of the summary and the abstract to clarify the aims of the paper

The title of the manuscript should be more specific reporting some words referred to the aim of the study

We have reworded the title of the paper to clarify the nature of the what is being presented.

The authors give great evidence to the concept of the drug resistance in tumor due to the presence of a cancer stem cell (CSC) subpopulation not responder to cytotoxic agents and potentially targetable with Cantrixil. The results showed in figure 7 are very interesting and give force to such thesis but the analysis was done only on three responder patients. The authors should comment in the results and in the discussion section of the manuscript the other possible causes of drug resistance in not responder or partially responder patients to Cantrixil.

As this is the first-in-human study with Cantrixil, many questions are still to be answered. Further studies are planned to better define the mechanism of action and the mechanisms of resistance to Cantrixil. We have made a number of modifications to the discussion and conclusion to address the comments raised. 

Reviewer 3 Report

In this phase I study the authors tested the safety and efficacy of a potential CSC targeting agent in patients with refractory/resistant OVCa.

The study was well design and the authros defined in a quick part A of the study the dose to be expanded in a subsequent cohort.

The use of drugs potentially targeting platinum resistant CSC is deifnitely warranted. Tere are data showing the potential of other less toxic drugs such as metformin to reduce the CSC in differnt tumors including OVCa.

In this study the dose of 5 mg/Kg showed substantial Grade 3 toxicity that has been observed at doses of 10 and 20 as well. The authors mentioned that the drug was active in preclinical models at high doses. Even if difficult to assess, the adverse events reported at 5 mg/Kg seem not to be different to those found with higher doses that would probably have stronger activity. This point about the flat adverse events should be discussed (the authors touched this pont about the possible IP portal treatment associated toxicity).

Another point that should be discussed is about the potential of the combination in pre-resitant tumors. The concomitant treatment of platinum and Cantrixil should strongly delay the onset of resistance na dlikely to have a more efficacious effect on OS.

The major limitation remains the toxicity of the treatment that resulted in discontinuation even at low doses.

Author Response

Please see the attached word document and also comments below:

The use of drugs potentially targeting platinum resistant CSC is deifnitely warranted. Tere are data showing the potential of other less toxic drugs such as metformin to reduce the CSC in differnt tumors including OVCa.

Preliminary data with metformin demonstrate that it may have a role. However, as with many other therapies in this patient population, the data available to date has been inconsistent and hampered by a lack of randomized controlled trials [Ahmed et al, 2021] We have not included metformin in our discussion because the focus of our discussion has been centered around the role of novel therapies given via IP administration.

Ahmed MF, Kanaan G, Mostafa JA. The Role of Metformin in Ovarian Cancer: Does Metformin Increase Survival in Ovarian Neoplasm?. Cureus. 2021;13(2):e13100. Published 2021 Feb 3. doi:10.7759/cureus.13100

In this study the dose of 5 mg/Kg showed substantial Grade 3 toxicity that has been observed at doses of 10 and 20 as well. The authors mentioned that the drug was active in preclinical models at high doses. Even if difficult to assess, the adverse events reported at 5 mg/Kg seem not to be different to those found with higher doses that would probably have stronger activity. This point about the flat adverse events should be discussed (the authors touched this pont about the possible IP portal treatment associated toxicity).

The distinguishing feature of the toxicity profile from the MTD determination has been that the patients treated with 10 and 20 mg/kg doses experienced Grade 3 ileus, this did not occur at the lower dose: hence the MTD of 5 mg/kg. Notwithstanding this, we agree with the reviewer that further exploration of the cause of the Grade 3 abdominal pain is required to better define the toxicity profile of Cantrixil.   

Another point that should be discussed is about the potential of the combination in pre-resitant tumors. The concomitant treatment of platinum and Cantrixil should strongly delay the onset of resistance na dlikely to have a more efficacious effect on OS.

We agree with the reviewer, and we hope to be able to evaluate the use of Cantrixil up-front in a future neoadjuvant study. As protocols for future studies are currently only being drafted, we are unable to provide details in this paper. Accordingly, the conclusion lists potential future avenues of study without providing details.

Round 2

Reviewer 1 Report

The manuscript has improved through the revision.

However, still the methodologies used to examine the changes in number and clonogenicity of circulating epithelial tumor cells and expression of the stem cell markers CD44 and ALDH have not been described. This must be added.

In addition, a discussion of the MTD and PK findings should be added.

Author Response

We thank the reviewer for their further comments and not that the following changes have been made to the manuscript. 

Materials and Methods:
An addition section has been added to the methods to accommodate this request , please see “2.6 Exploratory analysis of stem cell markers”.

Discussion:
We have included additional discussion of the PK results in the following paragraphs:

Discussion paragraph 1: “characterized by multi-exponential pharmacokinetics, with a rapid increase in systemic concentrations after the end of the infusion, followed by a rapid distributional phase and a slower elimination phase,”

Discussion paragraph 4: “Comparison of dose-normalized data indicates that Cantrixil does not display any clear lack of dose proportionality over the doses tested.”

We have included additional discussion of the MTD findings in the following paragraphs:

Discussion paragraph 3: “Overall, 16 (64%) patients experienced at least one grade 3 gastrointestinal adverse event. Of these, 3 patients consequently withdrew from the study; one due to the dose limiting toxicity of ileus/abdominal pain at a dose of 20 mg/kg, one due to bowel obstruction at a dose of 0.24 mg/kg and one due to a small intestinal obstruction as a dose of 5.0 mg/kg. While the MTD of 5.0 mg/kg was established on the basis the gastrointestinal findings, it is difficult to delineate whether the most frequently reported adverse events during Cantrixil monotherapy (vomiting, nausea, abdominal pain and bowel obstruction) arose from the Cantrixil itself or were a consequence of the IP port. Further exploration of the cause of the grade 3 abdominal pain is required to better define the toxicity profile of Cantrixil.”

Reviewer 3 Report

While i appreciate the temptative reply of the authors, i am concerned about the elusive way to answer, The point of the discontinuation has been absolutely not touched (and the others only marginally)

Author Response

We are surprised at this reviewer’s comments, given the favourable first round review, and had not intended to be elusive in our response. It is simply a case of not having sufficient data to draw definitive conclusions, hence the need for further studies.

We have added additional information about patients withdrawn from the study due to GI adverse events (see response to reviewer 1, round 2). Interpretation of the safety data are limited by the small sample size of this phase 1 study. It is for this reason that we have commented on the need for further studies at numerous points within the discussion. Hopefully future research in a larger population can be directed to answer these important clinical questions.